# Multi-Level Biomarkers for Early Diagnosis of Ischaemic Stroke: A Systematic Review and Meta-Analysis

**DOI:** 10.3390/ijms241813821

**Published:** 2023-09-07

**Authors:** Qianyun Li, Lingyun Zhao, Ching Long Chan, Yilin Zhang, See Wai Tong, Xiaodan Zhang, Joshua Wing Kei Ho, Yaqing Jiao, Timothy Hudson Rainer

**Affiliations:** 1Department of Emergency Medicine, University of Hong Kong, Hong Kong, China; u3008308@connect.hku.hk (Q.L.);; 2School of Biomedical Sciences, University of Hong Kong, Hong Kong, China

**Keywords:** biomarkers, ischaemic stroke, meta-analysis, diagnosis, systematic review

## Abstract

Blood biomarkers hold potential for the early diagnosis of ischaemic stroke (IS). We aimed to evaluate the current weight of evidence and identify potential biomarkers and biological pathways for further investigation. We searched PubMed, EMBASE, the Cochrane Library and Web of Science, used R package meta4diag for diagnostic meta-analysis and applied Gene Ontology (GO) analysis to identify vital biological processes (BPs). Among 8544 studies, we included 182 articles with a total of 30,446 participants: 15675 IS, 2317 haemorrhagic stroke (HS), 1798 stroke mimics, 846 transient ischaemic attack and 9810 control subjects. There were 518 pooled biomarkers including 203 proteins, 114 genes, 108 metabolites and 88 transcripts. Our study generated two shortlists of biomarkers for future research: one with optimal diagnostic performance and another with low selection bias. Glial fibrillary acidic protein was eligible for diagnostic meta-analysis, with summary sensitivities and specificities for differentiating HS from IS between 3 h and 24 h after stroke onset ranging from 73% to 80% and 77% to 97%, respectively. GO analysis revealed the top five BPs associated with IS. This study provides a holistic view of early diagnostic biomarkers in IS. Two shortlists of biomarkers and five BPs warrant future investigation.

## 1. Introduction

Stroke is a major cause of death and disability worldwide, with a profound impact on quality of life and a significant burden on families and society [1,2]. It results in 5.5 million deaths globally each year, with an estimated one in four adults experiencing a stroke in their lifetime [1,2]. Stroke can be classified into two main types: ischaemic stroke (IS) and haemorrhagic stroke (HS), with IS accounting for the majority of cases (71%) [2]. IS results from cerebral blood flow obstruction, and prompt reperfusion can significantly reduce mortality and disability [1]. Current reperfusion therapies include intravenous thrombolysis within 4.5 h (h) and endovascular thrombectomy within 6 h of symptom onset, which are time-critical [3]. Patients with evidence of salvageable brain tissue on imaging can also receive these therapies within 9 h for thrombolysis and 24 h for thrombectomy [3]. Whether patients within the time window can access either of these two therapies depends on the speed of differentiation of IS from HS and stroke mimics (SM). Currently, differentiation relies on clinical assessment and neuroimaging (brain CT/MRI). Non-contrast CT is used to exclude HS from thrombolysis treatment [4], but it is generally unavailable in prehospital settings and primary care hospitals. MRI is more sensitive for detecting IS, but fast access to it is restricted, and some patients with contraindications cannot undergo MRI scans [1]. Therefore, it is necessary to seek other approaches to compensate for the limitations of neuroimaging.

In recent years, with the advancement of point-of-care testing (POCT) technology, rapid testing has gradually become a reality for molecules ranging from DNA, RNA, proteins to metabolites, which could provide results within 5–20 min from blood sampling [5,6,7]. POCT devices are usually portable and user-friendly, leading to rapid expansion in applications across emergency departments, intensive care units, hospital wards, outpatient departments, and primary healthcare institutions [8,9,10]. If excellent diagnostic biomarkers of IS can be found, applying existing POCT technology would enable a rapid diagnosis of IS, benefiting both developing and developed regions. Therefore, a blood biomarker test has the potential to be a complement to neuroimaging and enhance the early management of IS.

A growing number of studies have investigated potential diagnostic biomarkers of IS. Previous systematic reviews of these studies have either evaluated single biomarkers [11] or single levels of biomarkers, such as proteins [12], RNAs [13], metabolites [14], and biomarker panels [15], separately. A comprehensive overview of multi-level biomarkers is necessary for a better assessment of the current evidence. High-throughput techniques have made it possible to evaluate vast quantities of molecules as biomarkers [16]. However, these omics studies have not yet been systematically analysed. Circulating biomarkers can also provide insights into pathophysiological changes related to brain injury [17]. Therefore, the aim of this study is: firstly, to identify all levels of early circulating diagnostic biomarkers of IS reported by published studies; secondly, to conduct a diagnostic meta-analysis to evaluate their pooled diagnostic performance if available; thirdly, to summarise all high-throughput omics studies on early diagnostic biomarkers of IS; fourthly, to apply bioinformatics to reveal pivotal biological processes (BPs) of IS.

We conducted a systematic search and analysis of studies on circulating biomarkers that distinguish IS from HS, SM, transient ischaemic attack (TIA), and controls within 24 h of symptom onset. We selected the 24 h window because diagnostic biomarkers during this early stage have the potential to expedite diagnosis and reperfusion treatment.

## 2. Methods

### 2.1. Design

We performed this systematic review and meta-analysis in compliance with the Preferred Reporting Items for Systematic Reviews and Meta-analyses (PRISMA) reporting guideline [18] and the Diagnostic Test Accuracy (DTA) extension [19]. The protocol was prospectively registered on PROSPERO (ID: CRD42022303870). Two reviewers independently performed screening, data extraction, and quality assessment. Disagreements were resolved by discussion between two reviewers (QL and one of YJ, CT, or LZ). We resorted to a third reviewer for unresolved conflicts.

### 2.2. Search Strategy and Selection Criteria

A literature search was conducted in PubMed (1971-), EMBASE (1974-), the Cochrane Library (1993-), and Web of Science (1900-). Our search included the following terms: “blood”, “plasma”, “serum”, “biomarkers”, “diagnosis”, “differentiation”, “stroke”, “ischaemic stroke”, “hemorrhagic stroke” and their corresponding synonyms. The detailed search strategies are presented in Appendix A. Searches were conducted on 4 June 2022. We imported all records from the literature search into Covidence systematic review software (Veritas Health Innovation, Melbourne, Australia, www.covidence.org, accessed on 29 September 2021), where screening, data extraction, and quality assessment were performed. The reference list from all the included studies and related reviews obtained from the screening process were also checked for any additional eligible articles.

Our inclusion criteria required all of five items: (1) studies that included at least two groups of the following: IS, HS, TIA, SM, and controls; (2) blood samples collected within 24 h from symptom onset; (3) any biomarkers or panel of biomarkers which included genes, RNAs, proteins, or metabolites; (4) patients aged ≥18 years; and (5) full-text articles available in English. Some studies reported data both within and exceeding 24 h, but we only extracted data that were within 24 h. We included diagnostic test studies, case control studies, and cohort studies. There were no restrictions on publication date. We excluded review articles, editorials, comments, and conference abstracts; articles reporting first blood samples collected longer than 24 h from symptom onset; biomarker levels tested in cerebrospinal fluid, urine, saliva, or breath; non-blood markers such as neuroimaging and clinical scores; and patients aged <18 years.

### 2.3. Data Extraction

We extracted general information (first author, publication year, title, country), characteristics of the study (study design, included groups, sample size, validation cohort, comparison, setting, reference standard, age, gender, specimen, sampling time, assay), biomarkers (omics name, panel or not, biomarker name, number of biomarkers, biological process), and outcomes (sensitivity, specificity, area under the curve (AUC), cut-off value, biomarker concentration, fold change). We also extracted true-positive (TP), false-positive (FP), false-negative (FN) and true-negative (TN) if data synthesis required. The extracted data were finally exported from Covidence into a csv spreadsheet.

### 2.4. Risk of Bias Assessment

We used Quality Assessment of Diagnostic Accuracy Studies (QUADAS-2) tool [20] to assess risk of bias and modified it for our study.

### 2.5. Statistical Analysis

We used R software (R 4.2.1, R core team) for qualitative and quantitative analysis. To exhibit the synthesised temporal change of biomarker concentration, we used a linear mixed model to acquire the combined median and 95% CI from multiple studies and then plotted them. To synthesise summary sensitivity and specificity, we used R package meta4diag, which implemented Bayesian bivariate meta-analysis of diagnostic test studies [21]. Compared to classical statistics, Bayesian statistics are more flexible and can better handle small sample sizes and unconventional data [22]. Bayesian bivariate meta-analysis considers the heterogeneity and correlation among different studies, as well as incorporating prior knowledge and uncertainty into the analysis, and providing more accurate estimation results and confidence intervals [22]. Furthermore, the meta4diag package utilises the newly proposed penalised complexity (PC) prior framework to enable users to specify prior distributions for the hyperparameters in an intuitive manner [21]. We used the default setting of meta4diag, which analysed a general diagnostic meta-analysis without detailed covariate information (e.g., study design and quality, patient characteristics, setting, person who performed the test, etc.) [21].

We performed the diagnostic meta-analysis when TP, FP, FN, or TN was available or could be calculated from a given sample size, sensitivity, and specificity in at least two studies on the same biomarker and at the same time point. Forest plots of summary sensitivity and specificity and summary receiver operating characteristic (SROC) curves were plotted. As a statistical inconsistency (e.g., I^2^) measurement was not typically applicable in meta-analysis of diagnostic test accuracy (DTA) studies, we replaced it in compliance with PRISMA-DTA by describing the term variability [19], which was the assessment of the similarity of assay methods, specimens, and cut-off values in this study. Meta-regression and sensitivity analysis were considered after variability assessment. A funnel plot was planned to evaluate publication bias. The GRADE (Grading of Recommendations Assessment, Development, and Evaluation) [23] system was considered for certainty assessment.

### 2.6. Bioinformatic Analysis

Gene Ontology (GO) enrichment analysis is a statistical method used to identify overrepresented BP, molecular functions, and cellular components within a set of genes or proteins [24,25]. The method is based on the GO, which is a standardised system of functional annotations for genes and proteins. GO enrichment analysis involves comparing a set of genes or proteins of interest to a background or reference set, such as all genes or proteins in a particular organism or cell type. The goal is to identify functional categories that are significantly overrepresented in the set of interest compared to the background set.

The diagnostic biomarkers of IS are typically identified by comparing the levels of specific molecules in patients with IS to those in other groups (HS/SM/TIA/controls). These biomarker molecules identified are likely to play a role in the BPs that occur after IS onset. By pooling all the protein and gene expression biomarkers from high-throughput omics studies, we performed GO enrichment analysis to determine which BPs are most likely to be implicated by these biomarkers. g:Profiler is a widely used web-based tool that is designed for performing GO analysis [26]. We input gene and protein lists in g:Profiler (accessed on 16 February 2023) which output the most significant BPs associated with the input biomarkers.

## 3. Results

### 3.1. Qualitative Synthesis

We screened 8544 records and assessed 426 full texts for eligibility. Finally, 182 studies were included (Figure 1), comprising IS (n = 15,675), HS (n = 2317), SM (n = 1798), TIA (n = 846), and controls (n = 9810), a total of 30,446 participants from 38 countries. The characteristics of the 182 included studies are presented in Appendix A. There were 89 prospective, 85 case control, 3 retrospective, and 5 cross-sectional studies. Most studies compared IS with controls (IS-control, n = 139), followed by IS-HS (n = 46), IS-SM (n = 17), and IS-TIA (n = 8). No study compared IS with all other four groups (HS, TIA, SM, and control) simultaneously. Half of the studies had a sample size less than 100 participants (n = 91), then 100 to 400 participants (n = 78), and more than 400 participants (n = 13). The majority of studies evaluated protein biomarkers (n = 127), followed by RNAs (n = 41), metabolites (n = 11), and gene expression (n = 9), whilst nine studies reported several levels of biomarkers simultaneously. A total of 19 (10%) studies had validation or replication cohorts to verify diagnostic performance of biomarkers in their study design.

A total of 518 biomarkers were pooled, including 203 proteins, 114 genes, 108 metabolites, and 88 transcripts. Transcript biomarkers consisted of 70 microRNAs, 7 circular-RNAs, 6 long-non-coding RNAs, 3 mRNAs, and 3 tRNAs. Of all pooled biomarkers, 427 biomarkers were from IS-controls, 135 from IS-HS, 36 from IS-SM, and 15 from IS-TIA comparisons (Figure 2A). The two overlapped biomarkers in this Venn diagram of IS versus all other conditions (HS/TIA/SM/control) were calcium-binding protein B (S100B) and Matrix metalloproteinase-9 (MMP-9), which means that they were reported to differentiate IS from all other conditions. Only a single assessment was conducted for 89% (444) of the biomarkers. Biomarkers that were evaluated four or more times are presented in Figure 2B, and they were the most frequently studied biomarkers. Brain-specific or brain-enriched biomarkers are listed in Appendix A.

We identified 72 studies (39.6%) that reported both sensitivity and specificity and acquired a list of biomarkers with the highest sensitivity and specificity (both over 90%) from these studies, which are presented in Table 1. The table also showed that these biomarkers have either not been validated or have only been validated in a few independent cohorts. Furthermore, we summarised 12 proteomics, 12 transcriptomics, 7 metabolomics, and 7 genomics high-throughput studies which are shown in Appendix A, and pooled 103 genomic, 79 proteomic, 41 transcriptomic, and 97 metabolomic shortlisted omics biomarkers with low selection bias, which are presented in Supplementary List S5. The shortlists from Table 1 and Supplementary List S5 are summarised for future validation.

### 3.2. Quality Assessment

Appendix A provides detailed results of our quality assessment using QUADAS-2. Of the 182 studies included in our analysis, 92 (51%) had a low risk of bias in patient selection, 180 (99%) had a low risk of index test, 145 (80%) had a low risk of bias in their reference standard, 146 (80%) had low risk in flow and timing, and 166 (91%) had low concern about patient not matching with the review question regarding prior testing, presentation, and setting. In addition, all included studies were found to have low concern that the biomarker test, its conduct, or interpretation differed from the review question. In total, 146 (80%) studies had low concern that the definition of stroke in the reference standard did not match the review question.

### 3.3. Quantitative Analysis and Meta-Analysis

Ten studies reported Glial Fibrillary Acidic Protein (GFAP) was eligible for performance of diagnostic meta-analysis (Appendix A). To investigate the temporal change in GFAP concentration, we additionally included two studies [47,48] for the calculation of a combined median with 95% CI. Our preliminary analysis indicated that patients with HS had higher circulating GFAP levels than those with IS at 3 h, 4.5 h, 6 h, and 24 h, reaching a peak at 4.5 h before gradually declining. The temporal trend is displayed in Figure 3.

Due to the limited number of available studies at each time point, with a maximum of six and a minimum of two, it was challenging to conduct a diagnostic meta-analysis. However, the R package meta4diag managed to mitigate these imperfections. As shown in the forest plot, the summary sensitivities and specificities of GFAP for detecting HS from IS were as follows: at 3 h, 73% (95%CI, 37–94%) and 97% (82–100%); at 4.5 h, 79% (62–91%) and 95% (83–100%); at 6 h, 80% (70–87%) and 92% (81–98%); and at 24 h, 71% (46–90%) and 77% (40–97%), respectively (see Figure 4). We also used SROC curves to evaluate the classification performance of GFAP for differentiating HS and IS at 3 h, 4.5 h, and 6 h (see Figure 5). The summary AUCs were 0.945 (95CI%, 0.641–0.999), 0.915 (0.835–0.996), and 0.849 (0.473–0.985) at 3 h, 4.5 h, and 6 h, respectively. However, we could not plot the SROC curve at 24 h due to the limited number of eligible studies (only two).

The level of variability (heterogeneity) among the synthesised studies at each time point was high, mainly due to differences in cut-off values, specimens, and assay methods. We were unable to perform a meta-regression to statistically confirm the cause of heterogeneity because there were too few studies available at each time point. Similarly, sensitivity analysis was not applicable since there were not enough synthesised studies after exclusion of the outlier studies. A funnel plot was also not applicable because of the small number of studies. Lastly, we could not apply GRADE due to the inapplicability of publication bias.

### 3.4. Biological Processes and GO Analysis

We extracted the BPs of proposed biomarkers reported in conventional non-high-throughput studies. The top five BPs were inflammation, oxidative stress, immune response, coagulation, and angiogenesis (see Figure 6A). We used GO analysis on gene and protein biomarkers from high-throughput omics studies and identified the top five BPs associated with IS, which were fibrinolysis, regulation of blood coagulation, regulation of haemostasis, blood coagulation, and haemostasis (see Figure 6B). The BPs acquired from conventional non-high-throughput studies and high-throughput studies were inconsistent.

## 4. Discussion

Our study presents a holistic view of early diagnostic circulating biomarkers in IS and summarises all high-throughput omics studies in this field for the first time. We analysed a total of 182 studies and pooled 518 multi-level biomarkers, which included genes, transcripts, proteins, and metabolites. We identified a shortlist of biomarkers with highest sensitivity and specificity, as well as a shortlist of high-throughput multilevel omics biomarkers with low selection bias for further validation. Moreover, we identified the top five vital BPs after IS onset. We also made a preliminary synthesis of the temporal change of GFAP within 24 h after stroke onset for the first time and synthesised summary sensitivity, specificity, and AUCs of GFAP in differentiating HS and IS according to different time points.

### 4.1. The Shortlist of Biomarkers with Optimal Diagnostic Performance

The shortlist of biomarkers with highest diagnostic performance is identified for further validation (Table 1). Biomarker research for IS remains wide-ranging and lacks depth, with the majority of biomarkers lacking sufficient independent validation cohorts. Although research teams frequently identify new potential biomarkers, the lack of adequate validation hinders their implementation in clinical settings. We list these biomarkers to encourage research teams to participate in their validation, improving their dependability and reproducibility. For example, as listed in Table 1, only three studies have investigated the circulating level of glycogen phosphorylase isoenzyme BB (GPBB) in the early stages of IS and two studies did not provide sensitivity and specificity. Therefore, rigorous validation is needed to evaluate the true value of these biomarkers.

### 4.2. The Synthesis of GFAP

GFAP is an intracellular class-III intermediate filament that is mainly expressed in the soma and end-feet of astrocytes throughout the brain [49]. After stroke onset, the blood–brain barrier (BBB) is disrupted, leading to the release of GFAP into the bloodstream [49]. Our tentative synthesis of GFAP circulating levels reveals that patients with HS have significantly higher levels of GFAP than those with IS all through the first 24 h. This difference may be due to the earlier disruption of BBB in patients with HS compared to those with IS. Our diagnostic meta-analysis of GFAP indicates that its diagnostic ability within 4.5 h is the best. However, this evidence of GFAP is inadequate for immediate clinical application. Nevertheless, GFAP could serve as an independent or complementary test for excluding HS from intravenous thrombolysis treatment, especially when CT is not routinely available [50]. More evidence is required to evaluate GFAP’s performance in differentiating IS from SM within 4.5 h time window. GFAP could also be a promising panel member when combined with other biomarkers, such as autoantibodies to NR2A/2B subunit of N-methyl-D-aspartate receptor (Abs to NR2), which achieved a sensitivity and specificity of 94% and 91% for differentiating IS from HS [45]. Our finding also suggests that the diagnostic performance of GFAP varies among different time points, indicating the importance of time points in evaluating the diagnostic capability of biomarkers.

### 4.3. High-Throughput Omics Studies

High-throughput omics studies are a promising approach to identifying stroke biomarkers, generating substantial data and information at various molecule levels [16]. In this regard, we systematically pooled these studies and the results are presented in Appendix A. We established a workflow for reference, starting with high-throughput methods to test gene expression/RNAs, proteins, or metabolites in the discovery stage. Differential molecules were then identified, and advanced methods such as machine learning/deep learning were applied to reduce the data dimension to a small number of molecules [43,51]. In the validation stage, shortlisted molecules were replicated/validated in a second or third independent cohort. Biomarkers obtained in this way had low selection bias and were often unimaginable from a conventional point of view. The cost of these studies was generally high, but they could provide new opportunities for the diagnosis and management of IS patients. To maximise the use of these findings, we pooled 79 proteins, 103 genes, 97 metabolites, and 41 RNAs omics biomarkers, which are listed in Supplementary List S5. This list of biomarkers could serve as a platform for future targeted omics studies, contributing to a better understanding of IS pathophysiology.

The integration of multi-omics studies will uncover comprehensive interaction networks between multiple molecular levels and provide a deeper insight into stroke pathophysiology and biomarker discovery [16]. Among the included omics studies, there was one multi-omics study that combined transcriptomics and proteomics together [52]. Moreover, advanced AI algorithms for integration analysis of multi-omics data are increasingly maturing. For instance, Multi-Omics Graph cOnvolutional NETworks (MOGONET), which allowed the joint exploration of omics-specific learning by graph convolutional networks and cross-omics correlation learning by view correlation discovery network, could identify more significant biomarkers from multiple omics data types [53]. Bringing together, understandings of IS will be moved forwards through emerging high techniques.

### 4.4. Pivotal Biological Processes

A thorough understanding of the pathophysiology of IS is essential for its optimal management [16]. In acute IS, the sudden decrease in cerebral blood flow initiates a series of cascade reactions and processes [54], leading to the alteration in thousands of molecules. Some of these molecules may serve as useful diagnostic biomarkers of IS [16]. Conversely, the reported diagnostic biomarkers might reveal the underlying BPs of IS. To identify the possible BPs implicated by these biomarkers, we conducted a GO enrichment analysis. This is the first systematic review of stroke biomarkers using GO analysis. We identified the top five enriched BP, including fibrinolysis, regulation of blood coagulation, regulation of haemostasis, blood coagulation, and haemostasis. These processes are consistent with our basic understanding of stroke pathology, confirming that they are critical pathophysiological processes that should be given greater attention when investigating diagnostic biomarkers and therapeutic targets. For instance, the commonly used alteplase for intravenous thrombolysis is a recombinant form of tissue plasminogen activator (t-PA) that mimics the endogenous t-PA involved in the natural fibrinolysis process [1]. Therefore, we speculate that the key diagnostic molecules of IS might exist in these processes.

### 4.5. Compare with Previous Related Studies

Previous systematic reviews on circulating diagnostic biomarkers of IS just assessed single biomarkers or single levels of biomarkers [11,12,13,14,15]. By contrast, we gathered genes, RNAs, proteins, and metabolite biomarkers together. There was a very recent systematic review which performed a meta-analysis for 25 protein biomarkers of IS based on their concentration [12]. However, synthesised concentration could not reveal the diagnostic performance and provide recommendations on clinical translation. Similarly, we also acquired a list of frequently studied protein biomarkers (Figure 2B) with reported concentrations that could be synthesised for conventional meta-analysis. However, we paid more attention to compare or synthesise diagnostic accuracy to find potential candidates. There were two previous diagnostic meta-analyses of GFAP, which mixed synthesised sensitivity and specificity of all time points together [55,56]. The difference was that our study distinguished multiple time points, as the circulating level of GFAP underwent obvious perturbation following time after stroke onset, which affected diagnostic performance substantially.

### 4.6. The Current Status and Recommendations for Future Research

Decades of research on hundreds of biomarkers for the early assessment of IS have not led to any translation, which might be due to the following flaws. Firstly, regarding study design, only half of the studies recruited consecutive patients prospectively. The most commonly used comparative group was healthy controls, which did not accurately reflect the clinical context. Only 9.3% of studies used a SM group, which is the most meaningful and challenging comparative group, even when MRI was available. Moreover, there were inconsistencies in settings, reference standards, sampling time points, specimens, and assays. A second issue was the small sample sizes in many studies, with 50% enrolling fewer than 100 participants. These sample sizes were too small to draw any convincing conclusions about the utility of the biomarker. The third concern was the challenge of comparing the diagnostic performance of the proposed biomarkers. Almost half (46.7%) of the studies only reported the concentration of biomarkers but without reporting sensitivity, specificity, or AUC. Most proposed biomarkers (89%) were only reported once, making comparisons impossible. There were indeed a few biomarkers that were reported more than once, and with reporting sensitivity and specificity, but they were measured at different time points, resulting in unavailable synthesis. Consequently, only the studies of GFAP was found to be available for synthesis, surprisingly at four time points: 3 h, 4.5 h, 12 h, and 24 h.

In future research, we recommend using prospective study designs, enrolling SM groups, establishing validation cohorts, and paying attention to dynamic changes in circulating biomarkers, with reporting items consistent with the START statement [57]. When exploring new biomarkers, studies should simultaneously validate or compare them with previously reported optimal biomarkers, enabling data synthesis and comparison to identify highly useful candidates. Researchers should aim to increase sample size through multi-centre cooperation and identify biomarkers for IS versus HS, SM, TIA, and healthy and patient controls. If blood biomarkers are to be used in accelerating diagnosis for reperfusion therapies, sampling time focused within 6 h of symptom onset would be more applicable. Furthermore, it is preferable to select molecules as candidate biomarkers that exhibit significant changes within the time window and have sufficient blood concentration to be detected.

## 5. Limitations

Firstly, we did not exclude studies with subarachnoid haemorrhage, as these few studies had a minor impact on the overall analysis. Secondly, we grouped all types of controls into a single category, including healthy controls, vascular risk factor controls, non-neurological disease controls, and other neurological disease controls. Thirdly, most studies included were not standardised diagnostic test studies, but we included them to gain an overview of all the potential biomarkers reported. Fourthly, considering the small quantity and great variability in the studies included in our diagnostic meta-analysis, it appears premature to perform a meta-analysis of GFAP, and more research is necessary to establish its actual efficacy in the future. Fifthly, although we conducted a careful quality assessment of the article, it is still possible that it may be influenced by some potential errors present in the original studies. These factors are beyond our control and cannot be addressed in this research.

## 6. Conclusions

The findings of this study have established a summarised platform for the future development of diagnostic biomarkers for IS. This platform includes a list of biomarkers with highest reported diagnostic performance, a list of unbiased omics biomarkers, potential BPs in which diagnostic biomarkers would most likely to be found, as well as an exploration of the issues existed in previous studies and future research directions. GFAP would be a potential biomarker for intravenous thrombolysis in patients with IS within 4.5 h from symptom onset at pre-hospital settings. These findings provide a foundation for future research that could have a significant impact on the augmentation of IS management.

## Figures and Tables

**Figure 1 ijms-24-13821-f001:**
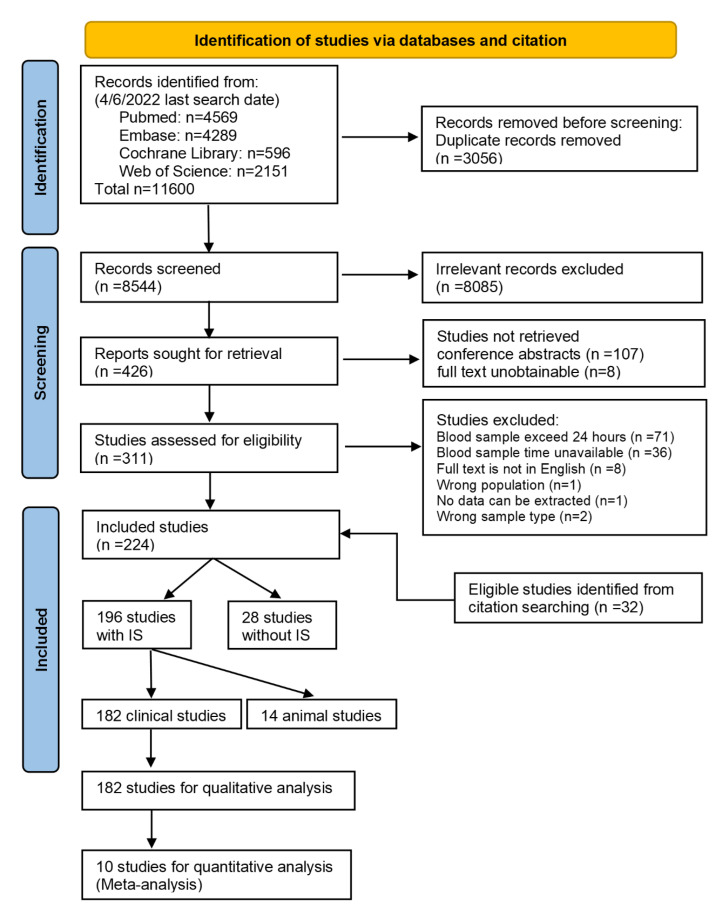
Flow diagram of search and selection process. IS, ischaemic stroke.

**Figure 2 ijms-24-13821-f002:**
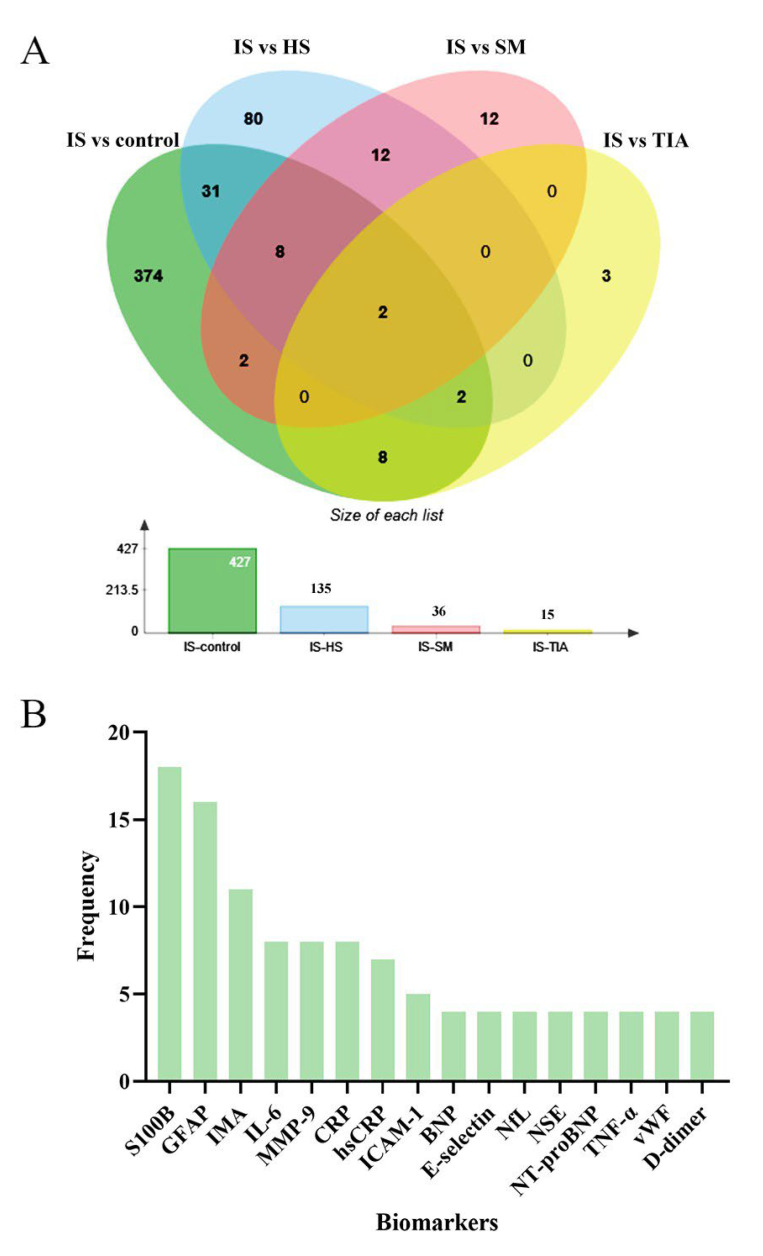
Venn chart for overlapped common biomarkers of IS versus control, HS, SM, and TIA groups (**A**) and most frequently studied biomarkers (**B**). A total of 427 biomarkers were pooled from IS-controls, 135 from IS-HS, 36 from IS-SM, and 15 from IS-TIA comparisons (**A**). The overlapped 2 common biomarkers were calcium-binding protein B (S100B) and Matrix metalloproteinase-9 (MMP-9). Overall, 16 biomarkers were studied ≥ 4 times among all studies included (**B**). S100B (n = 18); GFAP, glial fibrillary acidic protein (n = 16); IMA, ischaemia-modified albumin (n = 11); IL-6, interleukin 6 (n = 8); MMP-9, matrix metalloproteinase-9 (n = 8); CRP, C-reactive protein (n = 8); hsCRP, high-sensitive C-reactive protein (n = 7); ICAM-1, intercellular adhesion molecule 1 (n = 5); BNP, natriuretic peptides B (n = 4); NfL, neurofilament light chain (n = 4); NSE, neuron-specific enolase (n = 4); NT-proBNP, NT-pro-natriuretic peptides B (n = 4); TNF-α, tumour necrosis factor α (n = 4); vWF: von Willebrand factor (n = 4). IS, ischaemic stroke; HS, haemorrhagic stroke; SM, stroke mimics; TIA, transient ischaemic attack.

**Figure 3 ijms-24-13821-f003:**
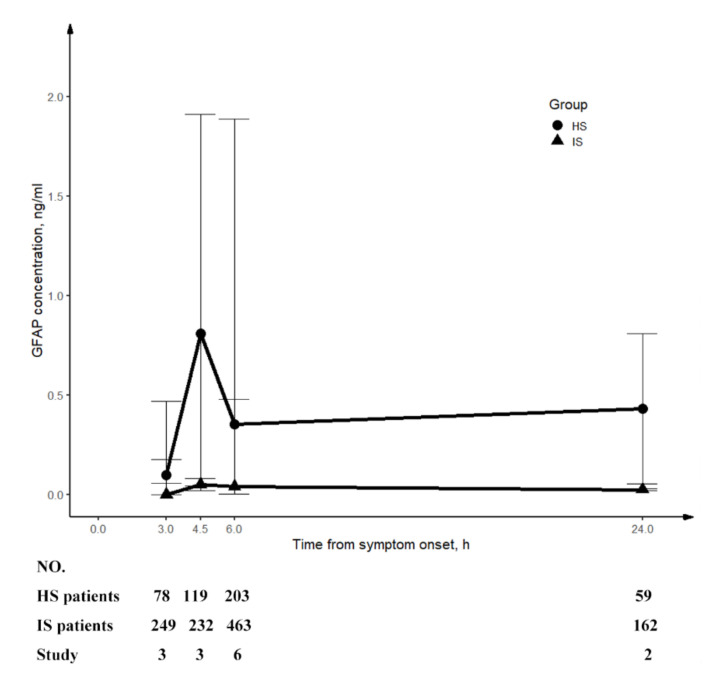
Temporal change in circulating level of GFAP. The concentration of GFAP (combined median with 95% CI) was plotted. This graph shows that GFAP increased rapidly from 3 h and peaked at 4.5 h from symptom onset and decreased continuously afterwards till 24 h in patients with HS. For patients with IS, GFAP had always been at a lower level than patients with HS. IS, ischaemic stroke; HS, haemorrhagic stroke; GFAP, glial fibrillary acidic protein.

**Figure 4 ijms-24-13821-f004:**
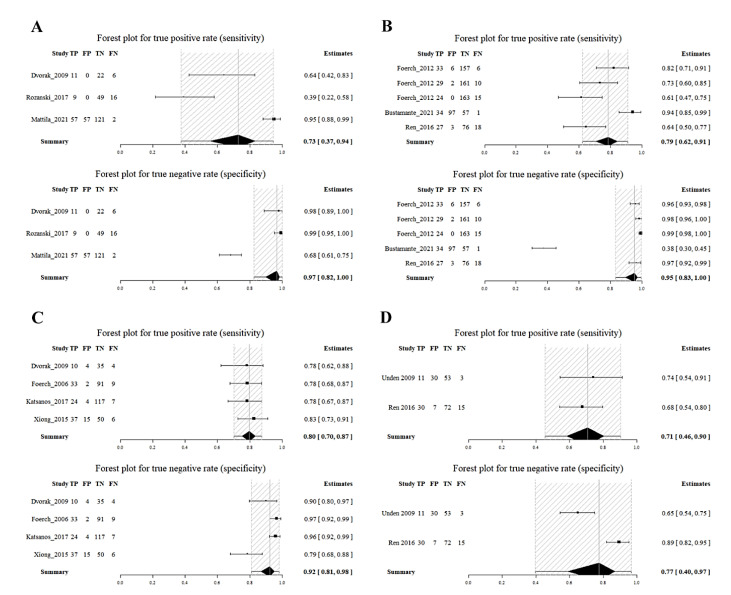
Summary sensitivity and specificity of GFAP at 3 h (**A**), 4.5 h (**B**), 6 h (**C**) and 24 h (**D**) in differentiating HS from IS. Summary sensitivity and specificity were 73% (95%CI, 37–94%) and 97% (82–100%) at 3 h; 79% (62–91%) and 95% (83–100%) at 4.5 h; 80% (70–87%) and 92% (81–98%) at 6 h; 71% (46–90%) and 77% (40–97%) at 24 h for differentiating HS from IS, respectively. IS, ischaemic stroke; HS, haemorrhagic stroke; GFAP, glial fibrillary acidic protein.

**Figure 5 ijms-24-13821-f005:**
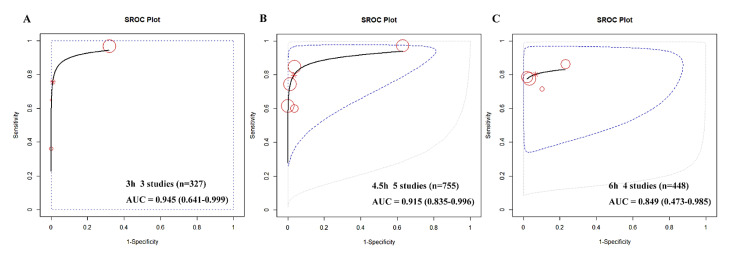
SROC of GFAP in differentiating HS from IS at 3 h (**A**), 4.5 h (**B**) and 6 h (**C**). The summary AUCs were 0.945 (95% CI, 0.641–0.999), 0.915 (0.835–0.996), 0.849 (0.473–0.985) at 3 h, 4.5 h, 6 h, respectively. Each circle represents a study, with larger circles indicating larger sample sizes and higher weights. SROC, summary receiver operating characteristic; AUC, area under curve; GFAP, glial fibrillary acidic protein; IS, ischaemic stroke; HS, haemorrhagic stroke.

**Figure 6 ijms-24-13821-f006:**
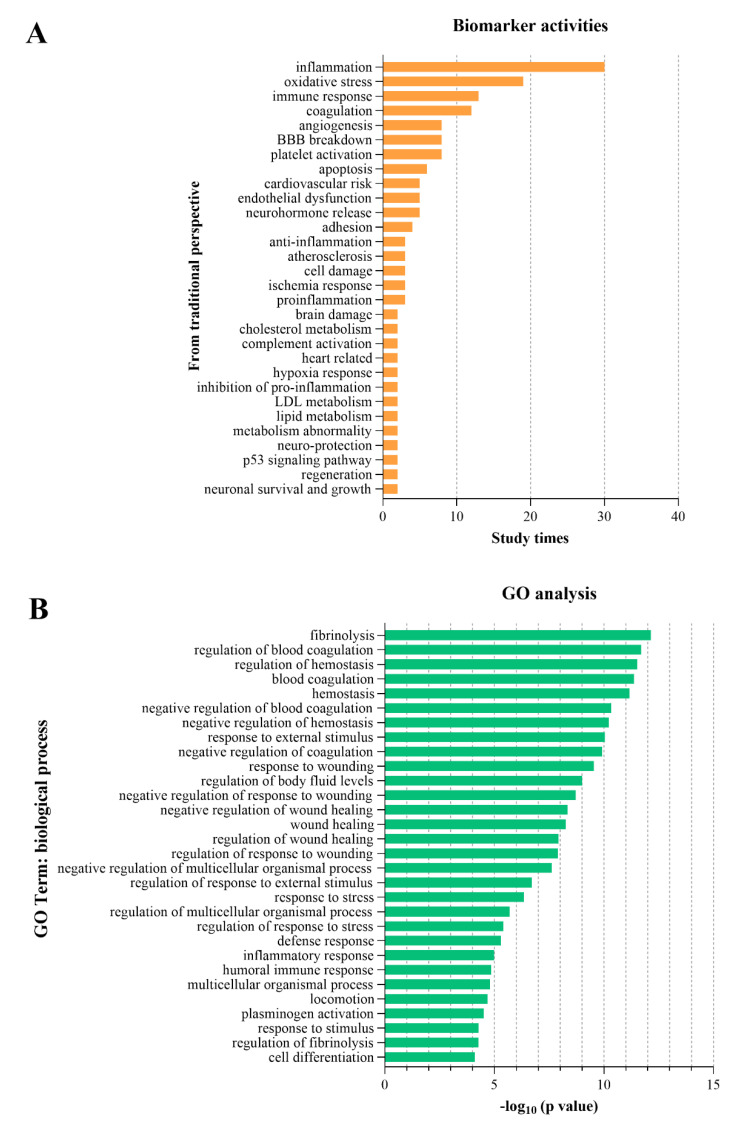
Top 30 biological processes reported from a conventional perspective (**A**) and from perspective of GO analysis of omics biomarkers (**B**). The top five BPs from conventional perspective were inflammation, oxidative stress, immune response, coagulation, and angiogenesis (**A**). GO analysis showed the top 5 pivotal BPs of ischaemic stroke from other conditions which were fibrinolysis, regulation of blood coagulation, regulation of haemostasis, blood coagulation, and haemostasis (**B**). BP: biological processes.

**Table 1 ijms-24-13821-t001:** Reported biomarkers with highest sensitivity and specificity for diagnosis of ischaemic stroke in early stage.

**Biomarkers**	**Study ID**	**Levels**	**Sampling Time**	**Sensitivity**	**Specificity**	**AUC**	**Comparison** **Groups**		**Sample Size**	**Biological** **Process**
Antibody to NR 2A/2B	Dambinova 2003 [27]	Protein	3 h	97%	98%	0.99	IS 1	HC 230 + VC 25 + ICH 18	304	Immune response
BDNF	Algin 2019 [28]	Protein	4 h	100%	92%	0.983	IS 75	HC 28	103	Neuronal survival and growth
IMA index	Ahn 2011 [29]	Protein	6 h	95.8%	96.4%	0.99	IS 28	SM 24	52	Ischemia response
NR2 peptide	Dambinova 2012 [30]	Protein	12 h	92.1%	96.5%	0.92	IS 50	SM 91 + VC 48 + HC 52	241	Brain cell damage
GPBB	Park 2018 [31]	Protein	12 h	93%	93%	0.96	IS 172	NSC 133	305	Ischaemia response
	Algawwam 2021 [32]	Protein	24 h	N/A	N/A	N/A	IS 40	HC 40	80	
ADAMTS13	Sharma 2015 [33]	Proteomics	24 h	90%	98%	0.96	IS 50	HC 35	85	Blood haemostasis and endothelial function regulation
S100A7	Sharma 2015 [33]	Proteomics	24 h	97%	91%	0.912	IS 50	HC 35	85	Blood haemostasis and endothelial function regulation
VILIP-1	Stejskal 2011 [34]	Protein	3 h	100%	100%	1.0	IS 16	HC 17	33	Brain cell damage
	Laterza 2006 [35]	Protein	24 h	N/A	N/A	N/A	IS 18	HC N/A	18	
	Algin 2019 [28]	Protein	4 h	No statistical significance			IS 75	HC 28	103	
miR-107	Yang 2016 [36]	Transcript	24 h	93.8%	92.2%	0.97	IS 114	HC58	172	Cerebral ischaemic injury
miR-124	Zhou 2021 [37]	Transcript	24 h	91.67%	93.52%	0.9527	IS 108	HC 108	216	Anti-inflammation
	Ji 2016 [38]	Transcript	24 h	N/A	N/A	0.69	IS 65	NSC 66	131	
	Liu 2015 [39]	Transcript	24 h	N/A	N/A	0.76	IS 31	HC 11	42	
APOA1-UP	Zhao 2016 [40]	Protein	24 h	90.3%	97.14%	0.975	IS 168	HC 104	272	Cholesterol metabolism
lncRNAs LINK-A	Ewida 2021 [41]	Transcript	24 h	92%	94%	0.914	IS 50	HS 25	75	Angiogenesis
*ANTXR2 + STK3 + PDK4 + CD163 + MAL + GRAP +* *ID3 + KIF1B +* *PLXDC2 + CTSZ*	O’Connell 2017 [42]	Gene	5 h, 24 h	95.7%	95.7%	0.997	IS 23	VC 23	46	Immune response
	O’Connell 2017 [42]	Gene	3 h	91.3%	95.7%	0.991	IS 23	VC 23	46	
	O’Connell 2016 [43]	Genomics	5.3 h	97.4%	100%	N/A	IS 39	NAC 24	63	
	O’Connell 2016 [43]	Genomics	4.6 h	92.3%	100%	N/A	IS 39	NAC 24	63	
	O’Connell 2016 [43]	Genomics	4.6 h	97.4%	90%	N/A	IS 39	SM 20	59	
S100B + BNGF + vWF + MMP9 + MCP-1	Reynolds 2003 [44]	Protein	3 h	98.1%	91.7%	N/A	IS 82	HS 103 + HC 214 + TBI 38	437	Brain cell damage, neuronal survive, coagulation, inflammation
	Reynolds 2003 [44]	Protein	6 h	98.1%	93.1%	N/A	IS 82	HS 103 + HC 214 + TBI 38	437	
GFAP+ antibody against NR2	Stanca 2015 [45]	Protein	12 h	94%	91%	N/A	IS 49	HS 23	72	Brain cell damage, immune response
17-peptide-panel	O’Connell 2019 [46]	Proteomics	12 h	93.3%	90%	0.95	IS 19	HS 17	36	Immune response

aAbs to NR2A/2B, autoantibodies (aAbs) to NR2A/2B subunit of N-methyl-D-aspartate receptor; BDNF, brain-derived neurotrophic factor; IMA index, ischaemia-modified albumin index; NR2 peptide, NR2 subunit peptide of N-methyl-D-aspartate receptor; GPBB, glycogen phosphorylase isoenzyme BB; ADAMTS13, von Willebrand factor-cleaving protease; S100A7, S100 calcium-binding protein A7; VILIP-1, visinin-like protein 1; APOA1-UP, apolipoprotein A1-unique peptide; HC: healthy control; VC: vascular risk control; NSC: non-stroke control; NAC: neurologically asymptomatic control; IS: ischaemic stroke; HS: haemorrhagic stroke; SM: stroke mimics; TIA: transient ischaemic attack; TBI: traumatic brain injury.

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
