# Peer review of "Multi-Level Biomarkers for Early Diagnosis of Ischaemic Stroke: A Systematic Review and Meta-Analysis"

_ijms, 2023, doi:10.3390/ijms241813821_

Round 1

Reviewer 1 Report

This is an extremely interesting and well conducted work on early diagnostic circulating biomarkers in ischemic stroke.

But in no way, the dosage of these markers can be presented as an alternative to neuroimaging in the early diagnosis of cerebral stroke.

First of all, neuroimages in acute stroke do not only serve to rule between ischemia and hemorrhage, even within 4.5 hours of onset. They give an infinity of other information, useful for the proper management of the patient in the acute phase.Then, how long is it necessary to get the results of dosages? Are they available 24/24? a single context in which it could possibly be considered would be the LMIC countries.

Hence, even the conclusion is excessively presumptuous.

Reviewer 2 Report

This is a very large literature review. As such it is vulnerable to errors in the papers reviewed, and this should perhaps be briefly discussed.

The authors seem to assume that blood measures of various metabolites will be useful in clinical management of brain vascular disease, but there is evidence to support this rather remarkable belief. Decisions in clinical management of stroke patients are made in the first  few hours (<5) after onset, so any so-called biomarker must be evident in that tight window. The review does not address this specifically. Imaging is by far the best current "marker" and is likely to remain so.

The authors' so-called biomarkers are metabolic responses to brain vascular events of various types occurring after the event. The review is useful in this sense but not as a diagnostic exercise. It should be rewritten with this in mind. It has biological significance, perhaps.

Round 2

Reviewer 1 Report

Please modify the last sentence of conclusion as follow: hese findings pro- vide a foundation for future research that could have a significant impact on the IS management.

OK for other changes.